# The Childhood Resilience Study: Resilience and emotional and behavioural wellbeing experienced by Australian Aboriginal and Torres Strait Islander boys and girls aged 5–9 years

**Deirdre Gartland**[1,2]*, **Arwen Nikolof**[2,3], **Fiona Mensah**[1,2], **Graham Gee**[1,4], **Karen Glover**[1,3], **Cathy Leane**[5], **Heather Carter**[6], **Stephanie Janne Brown**[1,2,3]

1 Murdoch Children's Research Institute, Intergenerational Health, Melbourne, Victoria, Australia, 2 Department of Paediatrics, University of Melbourne, Melbourne, Victoria, Australia, 3 South Australian Health and Medical Research Institute, Women's and Kids Theme, Adelaide, South Australia, Australia, 4 School of Psychological Sciences, University of Melbourne, Parkville, Victoria, Australia, 5 Women's and Children's Health Network, South Australia Health, Adelaide, South Australia, Australia, 6 Department for Education, Aboriginal Education Directorate, Adelaide, South Australia, Australia

* deirdre.gartland@mcri.edu.au

**Data Availability Statement:** No data are publicly available. Data cannot be shared publicly per the

## Abstract

### Background

Resilience is a process of drawing on internal or external strengths to regain, sustain or improve adaptive outcomes despite adversity. Using a child resilience measure co-designed with Aboriginal and Torres Strait Islander communities, we investigate: 1) children's personal, family, school and community strengths; 2) gender differences; and 3) associations between resilience and wellbeing.

### Methods

1132 parent/caregivers of children aged 5–12 years were recruited to the Childhood Resilience Study, including through the Aboriginal Families Study. The Aboriginal Families Study is a population-based cohort of 344 mothers of an Aboriginal and/or Torres Strait Islander child. This paper focuses on the wave 2 survey data on child resilience at age 5–9 years (n = 231). Resilience was assessed with the Child Resilience Questionnaire-parent/caregiver report (CRQ-P/C), categorised into tertiles of low, moderate and high scores. Child emotional/behavioural wellbeing and mental health competence was assessed with the parent-report Strengths and Difficulties Questionnaire. All Tobit regression models adjusted for child age.

### Outcomes

Aboriginal and Torres Strait Islander girls had higher resilience scores compared to boys (Adj.β = 0·9, 95%CI 0·9–1·4), with higher *School Engagement*, *Friends* and *Connectedness*

agreement between study investigators and the Aboriginal Health Council of South Australia to maximise participant privacy and confidentiality and protect Indigenous data sovereignty. Data sharing is subject to approval by the study''s Aboriginal Governance Group and Investigator team. Applications will be considered in context of papers in progress, compliance with conditions of ethics approval and consent, and potential to benefit Aboriginal communities. Interested researchers are invited to submit a request via the Aboriginal Families Study (afs@mcri.edu.au) or to contact Lead Investigators, Karen Glover (karen.glover@sahmri.com) and Stephanie Brown (stephanie.brown@mcri.edu.au).

**Funding:** The Aboriginal Families Study and Childhood Resilience Study were supported by three separate project grants from the Australian National Health and Medical Research Council (NHMRC) (#104395, #1105561, #1064061). Stephanie Brown is supported by NHMRC Leadership Investigator Grant (L2) #2018144. Arwen Nikolof is supported by an Australian Government Research Training Program PhD Scholarship and a top-up scholarship supported by the Royal Children's Hospital Research Foundation and Murdoch Children's Research Institute. Research conducted at the Murdoch Children's Research institute is supported by the Victorian Government's Operational Infrastructure program. The funders had no role in study design, data collection and analysis, decision to publish, or preparation of the manuscript.

**Competing interests:** The authors have declared that no competing interests exist.

*to language* scale scores. Resilience scores were strongly associated with wellbeing and high mental health competence. A higher proportion of girls with low resilience scores had positive wellbeing than did boys (73.3% versus 49.0%). High resilience scores were associated with lower SDQ total difficulties score after adjusting for child age, gender, maternal age and education and family location (major city, regional, remote) (Adj.β = -3.4, 95%CI -5.1, -1.7). Compared to the Childhood Resilience Study sample, Aboriginal Families Study children had higher mean CRQ-P/C scores in the personal and family domains.

## Interpretation

High family strengths can support Aboriginal and Torres Strait Islander children at both an individual and cultural level. Boys may benefit from added scaffolding by schools, family and communities to support their social and academic connectedness.

## Introduction

Child mental health is gaining increasing attention, with mental ill health being reported at younger ages and more frequently than previously [1,2]. In addition to the personal burden, poor child mental health can have developmental impacts and affect relationships within their family, peers and school, with longer term implications for personal and educational outcomes [3–5]. Much more is known about what drives poor mental health than what protects positive development and wellbeing, often described as resilience [6].

Historically, resilience has been described as the development of competence despite chronic stress [7], or positive developmental outcomes in the face of adversity [8]. More recent definitions encompass the potential for growth through adversity. From an Aboriginal standpoint, Kickett defines resilience as *"The ability to have a connection and belonging to one's land, family and culture: therefore an identity. Resilience allows the pain and suffering caused from adversities to heal. It is having a dreaming, where the past is brought to the present and present and the past are taken to the future. Resilience is a strong spirit that confronts and conquers racism and oppression strengthening the spirit. It is the ability not just to survive but to thrive in today's dominant culture [9]."* These descriptions encompass the two common factors frequently cited as necessary for defining resilience: first, the experience of adversity or stress, and second, the achievement of positive outcomes during or following the exposure to adversity. More recent definitions including the Kickett definition, also encompass the key role of culture, and the potential for growth through adversity. Accordingly, in this study resilience is defined as a socio-cultural and ecological oriented process where individuals have access to and draw on internal and external resources to "regain, sustain or improve" adaptive outcomes such as mental health or social and emotional wellbeing [6,9,10].

Studies investigating child resilience using validated measures are uncommon in both Indigenous and non-Indigenous contexts. There are few quantitative child measures available, and even fewer encompass both internal and external resources [11–13]. Recent reviews indicate the Strengths and Difficulties Questionnaire (SDQ), a measure of emotional and behavioural functioning or mental health, has most commonly been used to reflect resilience [11]. Such studies use positive functioning to identify resilience in the presence of an adversity such as family violence, whereas in this study, resilience is seen as the process that mediates between adversity and mental health or social and emotional wellbeing.

Whether individuals who experience adversity will have regained, sustained or improved mental health/wellbeing can depend on the types of adversity and contexts of these experiences. For example, adversity in Aboriginal and Torres Strait Islander communities is commonly linked to "*enduring legacies of colonization, continuous and cumulative transgenerational grief and loss, structural inequities, racism, and discrimination*" [12]. Aboriginal and Torres Strait Islander children can experience trauma directly or via secondary exposure–for example by bearing witness to the stories or traumatic experiences in the histories of their family and community members [14–16]. In this context, poor mental health or social/emotional wellbeing difficulties may be a more likely personal or community outcome, even when offset by resilience. Conversely, individuals with low resilience resources may have good mental health in the absence of adversity that challenges their internal or external vulnerabilities.

In response, we undertook the Childhood Resilience study, in which we co-designed a new multidomain measure–the Child Resilience Questionnaire–with Aboriginal/Torres Strait Islander and refugee-background communities [17,18]. Co-design originated from consultations with Aboriginal communities in urban, regional and remote areas of South Australia. Community members wanted to understand why some children were doing well, while others in similar situations were not seeming to do as well. We employed a strengths-based approach, with community consultation and bilateral knowledge exchange throughout, with Aboriginal research leadership (including authorship on this paper) and ongoing governance by an Aboriginal Governance Group for all aspects involving Aboriginal families. The Childhood Resilience Study involved over 300 Aboriginal and/or Torres Strait Islander parents/caregivers across the two stages of psychometric development [17]. The measure developed assesses the internal and external strengths and resources that can support a child in times of adversity [17].

In this paper, our focus is on parent/caregiver reports of child resilience in the Aboriginal Families Study, a population-based cohort of Aboriginal and Torres Strait Islander children aged 5–9 years, were used to investigate: 1) personal, family, school and community strengths of children taking part in the study; 2) gender differences in resilience scores; 3) the relationship between children's resilience scores and their emotional/behavioural wellbeing.

## Methods

### Setting and procedure

The Childhood Resilience Study was a five-year study to develop an inclusive, multidimensional measure of resilience for children that was relevant to a range of contexts in which children may encounter adversity. Study processes are described elsewhere [17]. Briefly, two methodological approaches ensured participation by families with diverse social and cultural backgrounds, adversity exposures and resilience factors: (1) co-design with Aboriginal and refugee background communities—two populations that have experienced high levels of historic and current discrimination, intergenerational trauma and varying forms of violence (e.g. structural racism); and (2) recruiting from a large public tertiary hospital. In Australia, public hospitals provide free healthcare and are attended by families with significant variation in where they live (rural/metropolitan), economic, cultural and social backgrounds [17]. In the final stage of the Childhood Resilience Study, child resilience data was collected with over 1000 families using the newly developed measure. Families were recruited between 26/9/2017 and 28/12/2020 from four sources: 1) a prospective, population-based cohort of Aboriginal and Torres Strait Islander families (Aboriginal Families Study); 2) four refugee background communities; 3) specialist clinic waiting rooms in a large tertiary children's hospital -); and 4) a prospective, population-based cohort of first-time mothers followed up over 10 years (Maternal Health Study).

**Consent.** Researchers, Aboriginal researchers and community researchers (refugee background communities) talked with potential families in person at homes, community spaces or events; or on the phone. They went through the study information statement in English or a preferred language (in refugee background communities). Families could ask questions before deciding to participate. Parents provided written or verbal consent for themselves and/or their child to participate. Where verbal consent was provided, the researcher completed a written consent form confirming that active verbal consent for participation had been gained. Children aged 7 or older were invited to provide verbal or written assent to self-completing the CRQ-C.

**Ethics approval.** The Childhood Resilience Study has been approved by the Royal Children's Hospital Human Research Ethics Committee (#34220, 36142, 37167) and the Aboriginal Health Research Ethics Committee, Aboriginal Health Council of South Australia (# 04-14-585) and the Department of Education and Training (2016_003144). The Aboriginal Families Study has been approved by the Royal Children's Hospital Human Research Ethics Committee (#36186) and the Aboriginal Health Research Ethics Committee, Aboriginal Health Council of South Australia (#04-16-689). Further details of Childhood Resilience Study processes and sample are available elsewhere [17]. This paper focuses on data collected in the Aboriginal Families Study (AFS), a prospective, population-based cohort of 344 Aboriginal and Torres Strait Islander children born July 2011-June 2013 in South Australia, and their mother/other primary caregiver. An Aboriginal Governance Group for the AFS was set up under the auspice of the Aboriginal Health Council of South Australia (AHCSA), the state's peak body for Aboriginal community-controlled health services in South Australia. This group had governance of the Childhood Resilience Study in relation to Aboriginal and Torres Strait Islander families and had a key role in the study design, conduct, interpretation of findings and outputs including publications. Recruitment and data collection in the AFS was conducted by Aboriginal researchers with family and community connections in urban, regional, and remote areas of South Australia. The Child Resilience Questionnaire-P/C (CRQ-P/C) was included in wave two, when children were starting primary school (24/01/2018 to 28/12/2020). Where children were not living with their mother, the child's primary caregiver was invited to complete a modified questionnaire. 231 mothers/caregivers participated and completed the CRQ-P/C. Further details are available in previous AFS papers.

| Panel 1. Child Reslience Questionnaire domains, scales and sample item. | |
| --- | --- |
| **PERSONAL** | |
| Self-Identity | My child is a strong person on the inside |
| Positive future | My child is positive about their life |
| Managing emotions | My child knows how to calm down when they feel angry |
| **FAMILY** | |
| Connectedness | My child talks to me about what is happening in their life |
| Basic needs | My child feels safe at our home |
| Family guidance | My child is given responsibilities in our family |
| **SCHOOL** | |
| Teacher support | My child has a teacher they can talk to when upset/angry |
| Engagement | My child likes learning at school |
| Friends | My child has a best/close friend |
| **COMMUNITY** | |
| Culture | My child is strong because of our family stories, values or spiritual beliefs |
| Language | My child can speak this language |

Additional information regarding the ethical, cultural, and scientific considerations specific to inclusivity in global research is included in the (S1 Checklist).

## Measures

**Resilience.** The Child Resilience Questionnaire parent/caregiver report (CRQ-P/C) [17] was co-designed with Aboriginal and refugee-background communities. The community-based collaborative development processes have been described elsewhere [17,18]. The 43-item measure has 11 scales across personal, family, school and community domains (see panel 1). Excellent to very good scale reliability has been observed for all but one scale with Cronbach alphas ranging from 0.88 (connectedness to language) to 0.73 (family guidance). The family basic needs scale had a Cronbach alpha of 0.61 indicating 'good' scale reliability [17]. Higher scores indicate the child has access to more strengths and resources when challenges arise. Parents/caregivers were asked "How often are the following true for your child: 0 "Not at all, 1 "Not often", 2 "Sometimes", 3 "Most of the time", 4 "All of the time". Mean scale, domain and total scores were calculated. Established cut off scores are not yet available, so the total CRQ-P/C score was divided into tertiles (thirds) to represent low, moderate, and high resilience scores respective to children within the sample.

**Emotional and/behavioural wellbeing.** The Strengths and Difficulties Questionnaire (SDQ) is a measure of behavioural and emotional difficulties for children aged 4–16 years. Participants rate 25 attributes as *not true*, *somewhat true* or *certainly true*. The total difficulties score derived from four subscales was used. Adopting a strengths-based approach, children with an SDQ total difficulties score below the threshold for high risk of emotional or behavioural problems (<17) were considered to have positive emotional/behavioural wellbeing [19,20].

O'Connor and colleagues describe using eight SDQ items to identify children who have very high mental health competence (MHC), who are 'flourishing' [21]. The items are: being considerate of other people's feelings; sharing readily with others; being helpful if someone is hurt, upset or feeling ill; being kind to younger children; volunteering to help others; being generally obedient; seeing tasks through to the end; and thinking things out before acting. The items capture key aspects of high-MHC. Item scores were summed (range 8–24) with a cut point of $\geq$23 indicating a very high level of MHC as described by the authors [21].

In addition, data were collected on a range of social factors including maternal age, postcode, household composition, education and employment. Child gender was reported by the caregiver. The Australian Bureau of Statistics Remoteness Structure was used to classify place of residence as urban, regional or remote based on relative access to services [22].

## Analyses

Tobit linear regression was used to examine differences in mean CRQ-P/C scale, domain and total scores by sample (AFS and larger Childhood Resilience Study sample) and AFS child gender as a more robust approach for censored data given the ceiling effects observed in scores [23]. Models adjusted for child age due to the significant developmental span from 5 to 9 years, and the potential for school resilience factors (and potentially others) to be affected by child age.

Small group numbers limited our capacity to examine child wellbeing (SDQ <17) by resilience tertiles and gender. Therefore, multi-variable Tobit linear regression models using the SDQ total difficulties score were undertaken. Models were a priori adjusted for mothers' age and highest educational qualification [24–26] and child age [26], as they are associated with SDQ scores.

## Results

### Aboriginal Families Study participants

The primary sample for this paper comprises the 231 Aboriginal Families Study (AFS) children with CRQ-P/C data–as reported by 208 mothers and 23 other primary caregivers. Social characteristics are presented in Table 1. Children ranged in age from five to nine years (with a mean age of 6·5, SD = 1·0). Of 23 children not living with their mother, four children were living with their father, 16 were living with grandparents/aunties, and three were in foster care. A third of the children lived in a single adult household. Over a third of the children spoke an Aboriginal and/or Torres Strait Islander language 'a little bit' (31·0%) or 'a lot' (7·9%), and over half of mothers spoke 'some words' of an Aboriginal and/or Torres Strait Islander language (55·7%) or fluently (6·4%).

There were no evident differences in the family characteristics for children with low, moderate or high resilience scores (see Table 1), with two exceptions. A higher proportion of the children whose mothers were not Aboriginal and/or Torres Strait Islander had a low resilience score (58·3%) compared to children of Aboriginal and/or Torres Strait Islander mothers (31·4%). Secondly, a higher proportion of children in families who did not have a government concession card (provided to low-income families) scored in the moderate resilience category (50.0%) compared to families who had the concession (29·3%). There were no differences in family characteristics of girls compared to boys (data available in S1 Table).

### Childhood Resilience Study participants (excluding AFS children)

For the 901 participants in the Childhood Resilience Study excluding the AFS children, it was primarily mothers (n = 943, 83.6%) or fathers (n = 152, 13.5%) completing the CRQ-P/C. Just under half the children were girls (n = 418, 47.7%) and they ranged in age from 5–13 years, with a mean age of 9.6 (SD = 2·1). The majority of children were born in Australia (n = 779, 87.0) and had 1–2 siblings (n = 613, 74.9%) or 3 or more (n = 183, 22.3%).

### Resilience in Aboriginal Families Study children compared to other children in the Childhood Resilience Study

Mean resilience scale and domain scores are presented in Fig 1 and Table 2 for the AFS children and other Childhood Resilience Study children. The mean scale scores were at the higher end of the range for most scales. As shown in Fig 1, there appeared to be ceiling effects which were most evident in the family domain.

Tobit regression models were used to examine differences in scale scores. The adjusted beta coefficient (β) represents the mean difference in CRQ-P/C scores for AFS children compared to other children in Childhood Resilience Study, adjusting for child age. In the personal domain, AFS children were scored higher by parents/caregivers on the *Identity* (Adj.β = 1·4, 95%CI 1·0, 1·9) and *Positive Future* scales (Adj.β = 0.9, 95%CI 0·3, 1·5) and on the *Personal Domain* overall (Adj.β = 0·5, 95%CI 0·1, 0·9). The children were also scored higher in the *Family Domain* (Adj.β = 0·4, 95%CI 0·1, 0·7), specifically on the *Connectedness* (Adj.β = 1·3, 95% CI 0·7, 1·8) and *Basic Needs* scales (Adj.β = 0·6, 95%CI 0·1, 1·1). The AFS children were scored lower on *Language Connectedness* (Adj.β = -1·3, 95%CI -1·8, -0.9) than other Childhood Resilience Study children (which included over 100 refugee background children) and on the community domain overall (Adj.β = -1·7, 95%CI -2·2, -1·2).

**Table 1. Social and family characteristics of children, mothers and other primary caregivers (n = 231).**

| | | Resilience Score (CRQ) | | | |
| | AFS | Low | Medium | High | |
| Characteristics | n (%) | n (%) | n (%) | n (%) | P value |
| All participants (n = 231) | | | | | |
| Participant | | | | | |
| Mother | 208 (90.0) | 66 (31.7) | 70 (33.7) | 72 (34.6) | |
| Other family member (e.g. father, aunty) | 20 (8.7) | 11 (55.0) | 6 (30.0) | 3 (15.0) | |
| Foster carer | 3 (1.3) | 2 (66.7) | 1 (33.3) | 0 (0.0) | 0.133 |
| Age | | | | | |
| 5–6 years | 129 (55.8) | 41 (31.8) | 40 (31.0) | 48 (37.2) | |
| 7–9 years | 102 (44.2) | 38 (37.3) | 37 (36.3) | 27 (26.5) | 0.224 |
| Participant reported gender | | | | | |
| Boy | 129 (55.8) | 52 (40.3) | 41 (31.8) | 36 (27.9) | |
| Girl | 102 (44.2) | 27 (26.5) | 36 (35.3) | 39 (38.2) | 0.072 |
| Place of residence (ABS classification 2016) | | | | | |
| Major city | 105 (45.5) | 34 (32.4) | 39 (37.1) | 32 (30.5) | |
| Regional | 81 (35.1) | 30 (37.0) | 23 (28.4) | 28 (34.6) | |
| Remote | 45 (19.5) | 15 (33.3) | 15 (33.3) | 15 (33.3) | 0.809 |
| Aboriginal and/or Torres Strait Islander mother | | | | | |
| Yes | 207 (89.6) | 65 (31.4) | 71 (34.3) | 71 (34.3) | |
| No[1] | 24 (10.4) | 14 (58.3) | 6 (25.0) | 4 (16.7) | 0.028 |
| Aboriginal and/or Torres Strait Islander father | | | | | |
| Yes | 175 (76.4) | 62 (35.4) | 53 (30.3) | 60 (34.3) | |
| No | 54 (23.6) | 17 (31.5) | 23 (42.6) | 14 (25.9) | 0.227 |
| Adults in household (past month) | | | | | |
| One adult | 73 (32.2) | 31 (42.5) | 17 (23.3) | 25 (34.2) | |
| Two adults | 115 (50.7) | 38 (33.0) | 42 (36.5) | 35 (30.4) | |
| 3+ adults | 39 (17.2) | 9 (23.1) | 16 (41.0) | 14 (35.9) | 0.163 |
| OWN children living with respondent | | | | | |
| None | 17 (7.4) | 9 (52.9) | 5 (29.4) | 3 (17.6) | |
| 1–2 | 92 (39.8) | 32 (34.8) | 36 (39.1) | 24 (26.1) | |
| 3–4 | 94 (40.7) | 26 (27.7) | 30 (31.9) | 38 (40.4) | |
| 5+ | 28 (12.1) | 12 (42.9) | 6 (21.4) | 10 (35.7) | 0.124 |
| OTHER children living with respondent | | | | | |
| None | 182 (78.8) | 62 (34.1) | 59 (32.4) | 61 (33.5) | |
| 1–2 | 35 (15.2) | 11 (31.4) | 15 (42.9) | 9 (25.7) | |
| 3+ | 14 (6.1) | 6 (42.9) | 3 (21.4) | 5 (35.7) | 0.631 |
| Age of participant | | | | | |
| 20–24 years | 12 (5.2) | 2 (16.7) | 3 (25.0) | 7 (58.3) | |
| 25–29 years | 75 (32.5) | 19 (25.3) | 31 (41.3) | 25 (33.3) | |
| 30–34 years | 71 (30.7) | 28 (39.4) | 21 (29.6) | 22 (31.0) | |
| 35+ years | 73 (31.6) | 30 (41.1) | 22 (30.1) | 21 (28.8) | 0.150 |
| Total | 231 (100) | 79 (34.2) | 77 (33.3) | 75 (32.5) | |
| Mothers[2] | | | | | |
| Mothers age at birth of study child | | | | | |
| 15–19 years | 32 (13.9) | 9 (28.1) | 12 (37.5) | 11 (34.4) | |
| 20–24 years | 82 (35.5) | 24 (29.3) | 28 (34.1) | 30 (36.6) | |
| 25–29 years | 62 (26.8) | 25 (40.3) | 21 (33.9) | 16 (25.8) | |

*(Continued)*

**Table 1.** (Continued)

| Characteristics | AFS | Resilience Score (CRQ) | | | |
| --- | --- | --- | --- | --- | --- |
| | | Low | Medium | High | |
| | n (%) | n (%) | n (%) | n (%) | P value |
| 30+ years | 55 (23.8) | 21 (38.2) | 16 (29.1) | 18 (32.7) | 0.722 |
| Relationship status | | | | | |
| Single | 95 (45.9) | 34 (35.8) | 25 (26.3) | 36 (37.9) | |
| Living with partner | 91 (44.0) | 22 (24.2) | 38 (41.8) | 31 (34.1) | |
| In a relationship, not living together | 21 (10.1) | 9 (42.9) | 7 (33.3) | 5 (23.8) | 0.123 |
| Highest level education | | | | | |
| Degree | 102 (50.0) | 31 (30.4) | 30 (29.4) | 41 (40.2) | |
| Certificate/Diploma | 31 (15.2) | 10 (32.3) | 9 (29.0) | 12 (38.7) | |
| Completed Year 12 | 56 (27.5) | 21 (37.5) | 21 (37.5) | 14 (25.0) | |
| Year 10 or less | 15 (7.4) | 4 (26.7) | 8 (53.3) | 3 (20.0) | 0.325 |
| Government concession card for low-income families | | | | | |
| No | 48 (23.1) | 11 (22.9) | 24 (50.0) | 13 (27.1) | |
| Yes | 160 (76.9) | 55 (34.4) | 46 (28.7) | 59 (36.9) | 0.023 |
| Currently employed (Full/Part Time) | | | | | |
| Yes | 78 (37.7) | 26 (33.3) | 31 (39.7) | 21 (26.9) | |
| No | 129 (62.3) | 40 (31.0) | 38 (29.5) | 51 (39.5) | 0.146 |
| Currently Studying (Full/Part Time) | | | | | |
| Yes | 36 (17.3) | 13 (36.1) | 16 (44.4) | 7 (19.4) | |
| No | 172 (82.7) | 53 (30.8) | 54 (31.4) | 65 (37.8) | 0.097 |
| Total | 208 (100) | 66 (31.7) | 70 (33.7) | 72 (34.6) | |

[1] Mothers of an Aboriginal and/or Torres strait Islander child.

[2] These details were not asked in the brief modified questionnaire completed by primary caregivers who were not the mother.

## Aboriginal Families Study children—Gender differences in resilience scores

Fig 2 and Table 3 present mean resilience scale, domain and total scores for AFS children by child gender. Girls had higher CRQ-P/C scores in all domains and on several subscales. Girls had a mean difference of up to one point higher on average compared to boys in the following domains: *Personal* (Adj.β = 1·0, 95%CI 0·1, 1·9); *School* (Adj. β = 1·2, 95%CI 0·1, 2·2); and *Community* (Adj.β = 0.7, 95%CI 0·1, 1·4); and overall on the *CRQ total score* (Adj. β = 0·9, 95% CI 0·3, 1·4).

On the subscales, gender differences were concentrated in the school domain where girls scored significantly higher on the *School Engagement* and *Friends* scales compared to boys (note: the *Friends* scale includes friends outside of school). There were similar numbers of girls (49·2%) and boys (50·8%) who spoke an Aboriginal and/or Torres Strait Islander language (a little bit or a lot), however, girls were estimated to score almost one point higher on average on the *Language Connectedness* scale (Adj. β = 0·9, 95%CI 0·4,1·5).

## Aboriginal Families Study children—resilience, wellbeing and mental health competence

Most children (80·3%) were classified as having 'positive wellbeing' (SDQ < 17). A slightly higher proportion of girls were classified as having positive wellbeing (85·2%) than boys (76·4%). The majority of those classified as not having positive wellbeing were boys (66·7%).

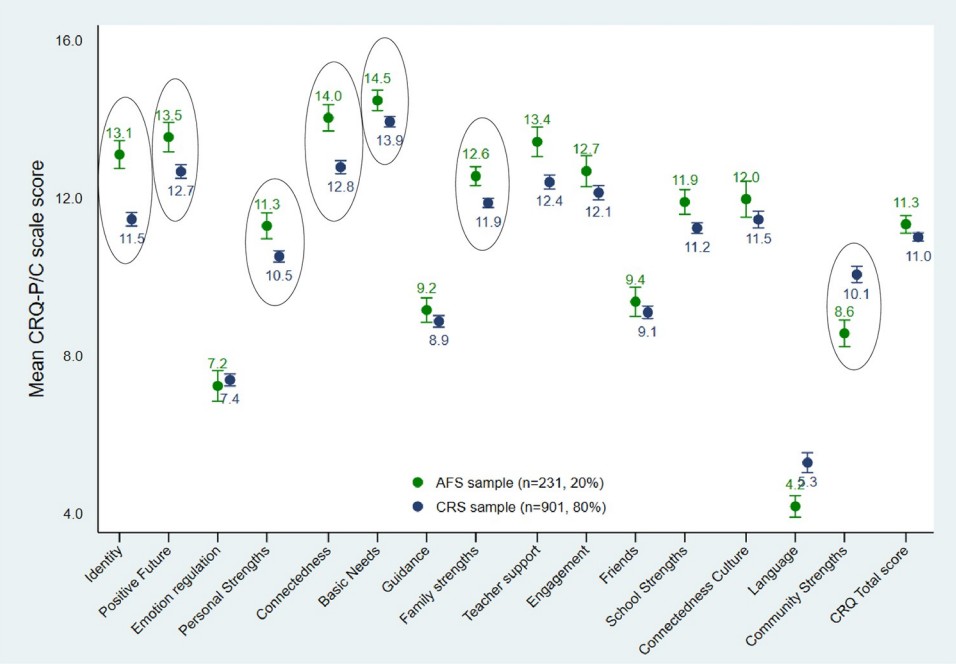

**Fig 1. Mean CRQ-P/C scale and domain scores with 95% confidence intervals for the Aboriginal Families Study (AFS) children and the Childhood Resilience Study children (excluding the AFS subsample).** (Circle = Tobit regression adjusting for child age is significant at p-value <0·05).

Almost one in five children (16·7%) had very high mental health competence (MHC) score. A higher proportion of girls (20·8%) were classified as having very high MHC than boys (13·4%).

The proportion of girls and boys with positive wellbeing (score <17 on SDQ) or high MHC by resilience score tertiles is presented in Fig 3. A larger proportion of children with high and medium resilience scores had positive wellbeing or very high MHC. The majority of girls with low resilience scores had positive wellbeing (70·4%). In contrast, boys with low resilience scores were almost equally divided between wellbeing (54·0%) and difficulties (46·0%). Of the children with low resilience scores, only 3·7% of girls and 2·0% of boys had very high MHC, in contrast around a third of girls and boys with high resilience scores had very high MHC.

To better understand the association between resilience scores, child gender and positive wellbeing, uni- and multi- variable models were conducted (see Table 4). In the univariable models, children with high resilience scores had emotional and/or behavioural difficulties scores 3.0 points lower on average (fewer difficulties) compared to children with moderate resilience scores (95%CI -4·7, -1·2). Children with low resilience scores had SDQ difficulties scores 4.4 points higher (more difficulties) on average compared to those with moderate resilience scores (95%CI 2·7, 6·1). Girls had lower SDQ difficulties scores on average compared to boys, and children living in remote areas had higher difficulties (see Table 4). No differences were observed for child age, mothers' age or higher education level.

In the fully adjusted multivariable model, children with low resilience scores had an average SDQ difficulty score 3.9 points higher than those with moderate scores (more difficulties). High resilience scores and female gender remained associated with significant decreases in the SDQ difficulties score (-3·4 and -1·3 respectively) after adjusting for child age, maternal age and education and location (see Table 4). In the adjusted model, lower maternal education was association with higher SDQ difficulties scores.

**Table 2. Mean CRQ-P/C domain and scale scores for the Aboriginal Families Study children by child gender, with Tobit logistic regression modelling differences between mean resilience scores for Aboriginal Families Study children compared to other Childhood Resilience Study children (n = 1132).**

| DOMAIN | | Childhood Resilience Study cohort (n = 1132) | Aboriginal Families Study sample (n = 231) | Childhood Resilience Study sample (not AFS) (n = 901) | Tobit Regression | |
|---|---|---|---|---|---|---|
| CRQ Scale | Items (range[1]) | Mean [95%CI] | Mean [95%CI] | Mean [95%CI] | Adj. β[2] [95% CI] | p-value |
| **PERSONAL strengths** | | | | | | |
| Self-identity | 4 (0–16) | 11.8 [11.6, 12.0] | 11.5 [11.3, 11.6] | 13.1 [12.8, 13.5] | 1.4 [1.0, 1.9] | <0.001 |
| Positive future | 4 (0–16) | 12.9 [12.7, 13.0] | 12.7 [12.5, 12.8] | 13.5 [13.2, 13.9] | 0.9 [0.3, 1.5] | 0.003 |
| Emotion regulation | 3 (0–12) | 7.4 [7.2, 7.5] | 7.4 [7.2, 7.5] | 7.2 [6.8, 7.6] | -0.1 [-0.5, 0.4] | 0.825 |
| *Mean personal domain score* | 11 (0–16) | 10.7 [10.5, 10.8] | 10.5 [10.4, 10.7] | 11.3 [11.0, 11.6] | 0.5 [0.1, 0.9] | 0.015 |
| **FAMILY strengths** | | | | | | |
| Connectedness | 4 (0–16) | 13.0 [12.9, 13.2] | 12.8 [12.6, 12.9] | 14.0 [13.7, 14.4] | 1.3 [0.7, 1.8] | <0.001 |
| Basic needs | 4 (0–16) | 14.0 [13.9, 14.2] | 13.9 [13.8, 14.1] | 14.5 [14.2, 14.7] | 0.6 [0.1, 1.1] | 0.022 |
| Guidance | 3 (0–12) | 8.9 [8.8, 9.1] | 8.9 [8.7, 9.0] | 9.2 [8.9, 9.5] | 0.2 [-0.2, 0.7] | 0.339 |
| *Mean family domain score* | 11 (0–16) | 12.0 [11.9, 12.1] | 11.9 [11.8, 12.0] | 12.6 [12.3, 12.8] | 0.4 [0.1, 0.7] | 0.021 |
| **SCHOOL strengths** | | | | | | |
| Teacher support | 4 (0–16) | 12.6 [12.5, 12.8] | 12.4 [12.2, 12.6] | 13.4 [13.1, 13.8] | 0.5 [-0.1, 1.1] | 0.086 |
| School engagement | 4 (0–16) | 12.2 [12.1, 12.4] | 12.1 [12.0, 12.3] | 12.7 [12.3, 13.1] | 0.2 [-0.4, 0.7] | 0.496 |
| Friends | 3 (0–12) | 9.2 [9.0, 9.3] | 9.1 [8.9, 9.3] | 9.4 [9.0, 9.7] | 0.4 [-0.1, 0.9] | 0.148 |
| *Mean school domain score* | 11 (0–16) | 11.4 [11.3, 11.5] | 11.2 [11.1, 11.4] | 11.9 [11.6, 12.2] | 0.2 [-0.2, 0.6] | 0.278 |
| **COMMUNITY strengths** | | | | | | |
| Cultural connectedness | 4 (0–16) | 11.6 [11.4, 11.8] | 11.5 [11.2, 11.7] | 12.0 [11.5, 12.4] | 0.2 [-0.4, 0.9] | 0.451 |
| Connectedness to language[3] | 4 (0–8) | 4.9 [4.7, 5.1] | 5.3 [5.0, 5.5] | 4.2 [3.9, 4.5] | -1.3 [-1.8, -0.9] | <0.001 |
| *Mean community domain score* | 8 (0–16) | 9.7 [9.6, 9.9] | 10.1 [9.9, 10.3] | 8.6 [8.2, 8.9] | -1.7 [-2.2, -1.2] | <0.001 |
| **Total RESILIENCE score** | 43 (0–16) | 11.1 [11.0, 11.2] | 11.0 [10.9, 11.1] | 11.3 [11.1, 11.6] | 0.0 [-0.3, 0.3] | 0.976 |

[1] Response options 0 'Not at all' to 4 'All of the time", except language where response options 0 'Not at all' to 2 'A lot"

[2] Models adjusted for child age

[3] Completed for multilingual children.

## Discussion

The Child Resilience Questionnaire offers a new quantitative approach to exploring resilience in children–describing personal strengths and strengths in their family, school and community. Use of participatory methods and codesign processes ensured content validity and development of a resilience measure that is culturally and socially inclusive. The measure showed good psychometric properties, with some evidence of ceiling effects. Child Resilience Questionnaire scores were strongly associated with positive wellbeing and high mental health competence for Aboriginal and Torres Strait Islander children at ages 5–9 years. Child personal and family strengths were scored higher than other children in the Childhood Resilience Study sample. Family strengths potentially support Aboriginal and Torres Strait Islander children at both an individual and cultural level. Overall, and on specific scales, Aboriginal girls were scored higher than boys, who may benefit from added scaffolding by schools, family and communities to support their social and academic connectedness.

Compared to other Childhood Resilience Study children, parents/caregivers in the Aboriginal Families Study scored their child 's resilience particularly high in terms of self-identity (e.g.

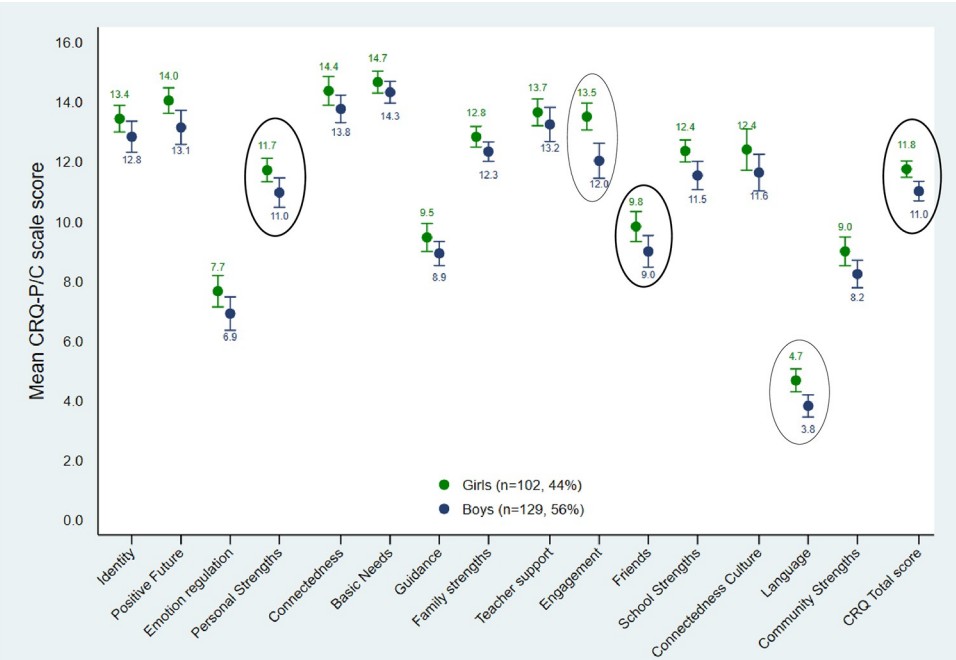

**Fig 2. Mean CRQ-P/C scale and domain scores with 95% confidence intervals by gender (n = 231).** (Circle = Tobit regression adjusting for child age is significant at p-value <0·05).

having self-confidence), positive sense of the future, family connectedness (e.g. someone the child can share their feelings with), and family basic needs (e.g. knowing they are loved). While most of the resilience research in Australia and internationally is focused on youth or adults, the importance of these particular strengths is reflected in the literature. For example, in one interview study, Aboriginal community members identified that the key to child resilience was "knowledge and self-belief that encouraged positive decision making despite challenges" [27]. A recent scoping review of Australian programs, processes, and practices to promote individual and/or collective Indigenous resilience or wellbeing reported that most of the eight publications identified emphasized the need for strategies to strengthen individual or community connection to culture [12]. Finally, a Canadian study utilising Indigenous methodologies reported that strengthening cultural identity and family connections was one of three intersecting processes that facilitate youth resilience and wellness [25]."

Aboriginal and Torres Strait Islander family structures are diverse and often involve sharing of responsibilities for bringing up children. This can include children spending time in more than one household, high levels of interaction with extended family and time spent with elders/community leaders. Parents may share direct care of their child over a period of time for voluntary (spending time with extended family) or involuntary reasons (including ongoing impacts of colonisation, the stolen generations, discrimination, structural disadvantage, and inequity) [14,28]. The high scores for children in the family domain are likely to reflect these more collective values and kinship structures [29,30]. A key role for many Aboriginal and Torres Strait Islander family members is to foster children's learning about their identity and culture [28,30,31]. Parents report that learning about culture supports their child's development in terms of cultural practice and behaviour, family and community connections and their self-identify [28,30,32]. Thus where more extensive family resources and supports are available to Aboriginal and Torres Strait Islander children, it may also be considered a personal **and**

**Table 3. Mean CRQ-P/C domain and scale scores for the Aboriginal Families Study children by child gender, with Tobit logistic regression modelling differences between mean resilience scores for girls compared to boys (n = 231).**

| DOMAIN | | Sample (n = 229) | Boys (n = 101) | Girls (n = 128) | Tobit Regression | |
|---|---|---|---|---|---|---|
| CRQ Scale | Items (range[1]) | Mean [95%CI] | Mean [95%CI] | Mean [95%CI] | Adj. β[2] [95%CI] | p-value |
| **PERSONAL strengths** | | | | | | |
| Self-identity | 4 (0–16) | 13.1 [12.8, 13.5] | 12.8 [12.3, 13.4] | 13.4 [13.0, 13.9] | 1.2 [-0.2, 2.6] | 0.085 |
| Positive future | 4 (0–16) | 13.5 [13.2, 13.9] | 13.1 [12.6, 13.7] | 14.0 [13.6, 14.5] | 2.4 [-0.1, 4.8] | 0.057 |
| Emotion regulation | 3 (0–12) | 7.2 [6.8, 7.6] | 6.9 [6.3, 7.5] | 7.7 [7.1, 8.2] | 0.8 [-0.1, 1.7] | 0.068 |
| *Mean personal domain score* | 11 (0–16) | 11.3 [11.0, 11.6] | 11.0 [10.5, 11.5] | 11.7 [11.3, 12.1] | 1.0 [0.1, 1.9] | 0.031 |
| **FAMILY strengths** | | | | | | |
| Connectedness | 4 (0–16) | 14.0 [13.7, 14.4] | 13.8 [13.3, 14.2] | 14.4 [13.9, 14.9] | 2.3 [0.0, 4.5] | 0.045 |
| Basic needs | 4 (0–16) | 14.5 [14.2, 14.7] | 14.3 [14.0, 14.7] | 14.7 [14.3, 15.0] | 0.3 [-1.6, 2.3] | 0.746 |
| Guidance | 3 (0–12) | 9.2 [8.9, 9.5] | 8.9 [8.5, 9.3] | 9.5 [9.0, 9.9] | 0.8 [0.0, 1.6] | 0.049 |
| *Mean family domain score* | 11 (0–16) | 12.6 [12.3, 12.8] | 12.3 [12.0, 12.7] | 12.8 [12.5, 13.2] | 0.5 [-0.3, 1.4] | 0.236 |
| **SCHOOL strengths** | | | | | | |
| Teacher support | 4 (0–16) | 13.4 [13.1, 13.8] | 13.2 [12.7, 13.8] | 13.7 [13.2, 14.1] | 1.7 [-0.2, 3.5] | 0.074 |
| School engagement | 4 (0–16) | 12.7 [12.3, 13.1] | 12.0 [11.4, 12.6] | 13.5 [13.1, 14.0] | 2.5 [1.0, 4.0] | 0.001 |
| Friends | 3 (0–12) | 9.4 [9.0, 9.7] | 9.0 [8.5, 9.5] | 9.8 [9.3, 10.3] | 1.1 [0.1, 2.1] | 0.028 |
| *Mean school domain score* | 11 (0–16) | 11.9 [11.6, 12.2] | 11.5 [11.1, 12.0] | 12.4 [12.0, 12.7] | 1.2 [0.1, 2.2] | 0.028 |
| **COMMUNITY strengths** | | | | | | |
| Cultural connectedness | 4 (0–16) | 12.0 [11.5, 12.4] | 11.6 [11.0, 12.2] | 12.4 [11.7, 13.1] | 0.7 [-0.7, 2.0] | 0.333 |
| Connectedness to language[2] | 4 (0–8) | 4.2 [3.9, 4.5] | 3.8 [3.4, 4.2] | 4.7 [4.3, 5.1] | 1.0 [0.4, 1.6] | 0.002 |
| *Mean community domain score* | 8 (0–16) | 8.6 [8.2, 8.9] | 8.2 [7.8, 8.7] | 9.0 [8.5, 9.5] | 0.7 [0.1, 1.4] | 0.029 |
| **Total RESILIENCE score** | 43 (0–16) | 11.3 [11.1, 11.6] | 11.0 [10.7, 11.3] | 11.8 [11.5, 12.0] | 0.9 [0.3, 1.4] | 0.005 |

[1] Response options 0 'Not at all' to 4 'All of the time", except language where response options 0 'Not at all' to 2 'A lot"

[2] Models adjusted for child age

[3] Completed for multilingual children.

cultural strength in this context. These strengths will support Aboriginal and Torres Strait Islander children in challenging times and provide vital social and cultural scaffolding for ongoing resilience as children grow and develop [33].

Caregivers reported lower resilience scores for boys in comparison to girls in terms of academic engagement and friendships. *Academic engagement* included items about liking school, being interested in what they learn and trying hard at school. The *Friends* scale included items on whether the child had a close friend, a friend with whom they could share their worries and a group of friends they have fun with. Whilst analyses adjusted for child age, these gendered findings may reflect general developmental educational and social trajectories. Aboriginal and Torres Strait Islander student outcomes also generally show higher scores for girls compared to boys [34]. Improving Aboriginal and Torres Strait Islander education outcomes (particularly in remote areas) has been a strong priority in national education policies, with a great deal of research, policy and program implementation focused on this issue–predominantly from a deficit perspective [34]. Some have argued that it is both a failing of an education system where the 'standardised' Australian curriculum is predominantly serving "white workforce interests", combined with a lack of culturally appropriate assessment tools [35,36]. Importantly, the complex intersectionality between race/ethnicity, gender and poverty is rarely addressed [36], with Aboriginal and Torres Strait Islander students having the added overlay of colonisation impacts (including racism, segregation, dispossession, and forced

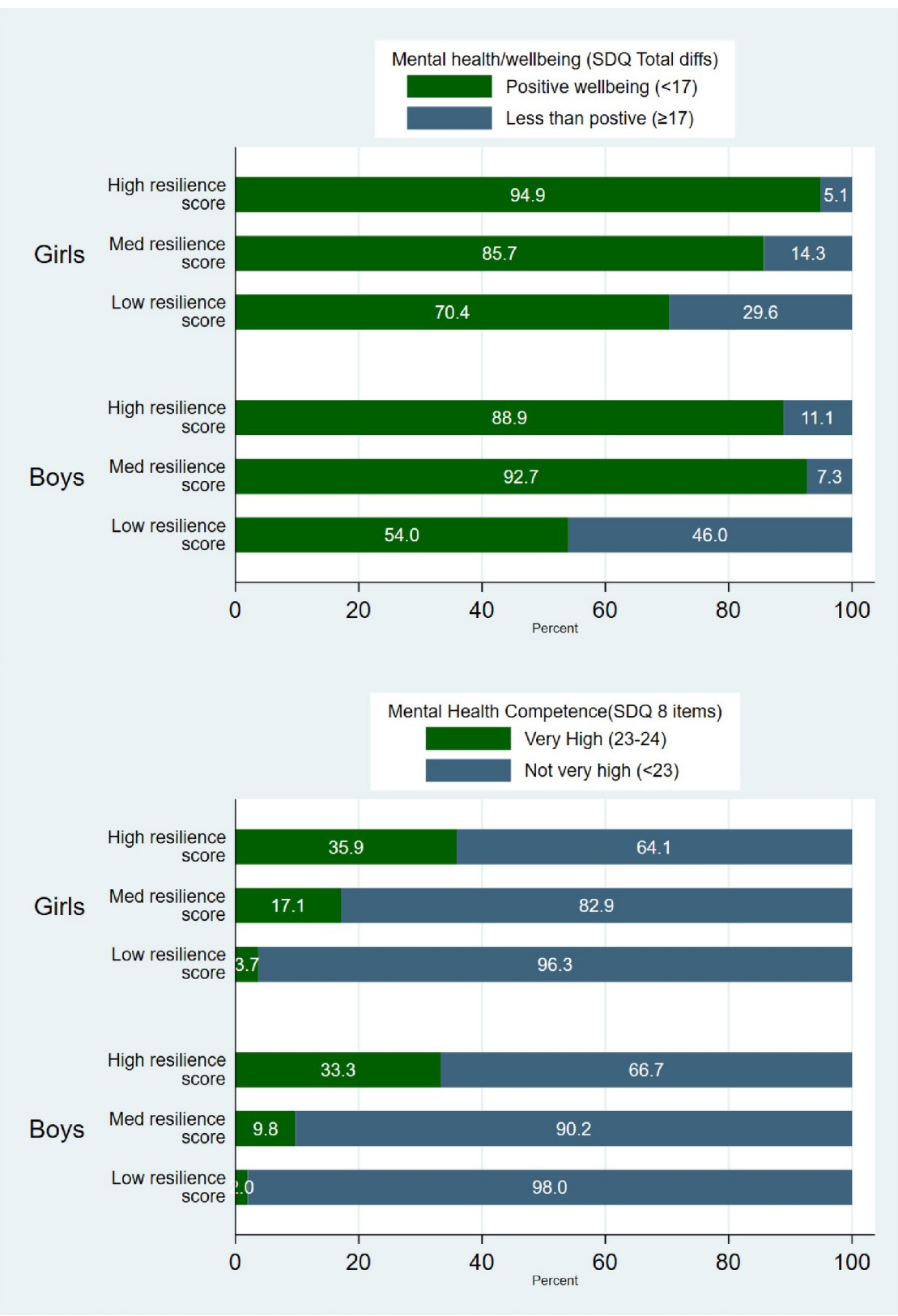

**Fig 3. Proportion of girls and boys in the Aboriginal Families Study with positive wellbeing or high mental health competence in each resilience category (high/medium and low scores) (n = 231).**

**Table 4. Univariable and multivariable linear regression models of child resilience and gender on the SDQ total difficulties score, adjusting for sociodemographic factors (n = 231).**

| Total difficulties score (SDQ) | Cohort | Univariable models | | Multivariable model | |
|---|---|---|---|---|---|
| | | β [95%CI] | p-value | Adj.β [95%CI] | p-value |
| Resilience scores (tertiles) | | | | | |
| High | 77 (33.3) | -3.0 [-4.7, -1.2] | 0·002 | -3.4 [-5.2, -1.7] | <0·001 |
| Moderate | 75 (32.5) | 1.0 [ref] | | 1.0 [ref] | |
| Low | 79 (34.2) | 4.4 [2.7, 6.1] | <0·001 | 3.9 [2.2, 5.7] | <0·001 |
| Child age | | | | | |
| 5–6 years | 129 (55.8) | 1.0 [ref] | | 1.0 [ref] | |
| 7–8 years | 102 (44.2) | 1.2 [-0.5–2.8] | | -0.1 [-1.6–1.4] | 0.901 |
| Child gender | | | | | |
| Male | 129 (55.8) | 1.0 [ref] | | 1.0 [ref] | |
| Female | 102 (44.2) | -2.6 [-4.2, -1.0] | 0·002 | -1.3 [-2.8, 0.1] | 0.063 |
| Location of residence (ABS 2016) | | | | | |
| Major city | 105 (45.5) | 1.0 [ref] | | 1.0 [ref] | |
| Regional | 81 (35.1) | 0.8 [-1.0, 2.6] | 0·314 | 0.8 [-0.8, 2.4] | 0.301 |
| Remote | 45 (19.5) | 2.5 [0.3, 4.7] | 0·023 | 1.7 [-0.3, 3.6] | 0·094 |
| Mothers' age | | | | | |
| 20–24 years | 71 (30.7) | -3.8 [-7.5, -0.0] | 0·051 | -2.5 [-5.7, 0.7] | 0.129 |
| 25–29 years | 12 (5.2) | -1.9 [-3.9, 0.1] | 0·061 | -1.2 [-3.0, 0.6] | 0.183 |
| 30–34 years | 75 (32.5) | 1.0 [ref] | | 1.0 [ref] | |
| 35+ years | 73 (31.6) | -0.7 [-2.7, 1.3] | 0·557 | -1.4 [-3.2, 0.4] | 0·133 |
| Mother's highest educational qualification | | | | | |
| Diploma/Certificate/Degree | 71 (34.8) | 1.0 [ref] | | 1.0 [ref] | |
| Year 12 or less | 133 (65.2) | 1.6 [-0.1, 3.4] | 0·067 | 2.4 [0.9, 4.0] | 0.002 |
| Intercept | | | | 9.6 [7.8, 11.9] | <0.001 |

disconnection from culture). Targeted school approaches to support Aboriginal and Torres Strait Islander students' academic and social connections are gaining greater attention, with schools that recognise poverty as a contributing factor showing some gains [37,38]. No comparative Australian or international research examining child resilience by gender was identified. A 2009 paper described an analysis conducted to explore gender differences in protective factors using the US Longitudinal Survey of Youth study Waves 3–6 (1986–1996) [39]. Notably, the findings apply to youth rather than children, and resilience was defined in terms of <u>not</u> having a negative outcome (rather than as a process as is used in this study). The authors reported that having a positive school environment (in secondary school) was associated with resilience against delinquency and drug use for girls but not for boys. Further research reflecting the current understanding of resilience as a process is imperative.

Over 70% of Aboriginal and Torres Strait Islander children in this study scored at low risk of emotional and/or behavioural difficulties on the SDQ (<17), almost identical to the 72% reported in the SEARCH study of Aboriginal children aged 4–17 years [20]. The proportion of boys and girls aged 6–7 years with very high mental health competence in this sample (16.7 and 22.3 respectively) was similar to that reported for the Longitudinal Study of Australian Children (LSAC, 14.4 and 26.0 respectively) [21]. While the proportion of children with emotional and/or behavioural difficulties did not differ for boys and girls with medium and high resilience scores, more boys with low resilience scores had difficulties compared to girls. Similarly, few boys with low resilience scores were classified as having high MHC. These are important findings suggesting that there may be more negative implications for boys who do not

have access to resources that can support them to navigate adversity. Multilevel early interventions focused on strengthening resilience factors such as connections with school and emotional/behavioural skills for Aboriginal and Torres Strait Islander boys are likely to benefit them across their life course. To be effective, such efforts must be culturally based and privilege local community knowledge and priorities. They must also sit within broader efforts to 1) foster healing and recovery from intergenerational and collective trauma; and 2) address the social inequities driving disadvantage in Aboriginal and Torres Strait Islander communities [30,33].

Our study is the first to provide evidence of an association between resilience and wellbeing for Aboriginal and Torres Strait Islander children in middle childhood: a key developmental stage where early intervention can have significant and lifelong impacts. Further research is required to establish whether the gender difference in resilience scores observed for Aboriginal and Torres Strait Islander children aged 7–9 years is sustained at older ages, and to explore resilience, gender differences and wellbeing in other populations. Other strengths of the study include co-design of the CRQ-P/C with Aboriginal and Torres Strait Islander communities in urban, regional and remote areas of South Australia, and study governance provided by members of the study's Aboriginal Governance Group and Aboriginal authors (KG, CL, AN, GG) of this paper. While the CRQ-P/C was designed for use with Aboriginal and Torres Strait Islander children, the utility of the SDQ for this population has been questioned in terms of cultural sensitivity/content and psychometric properties [40]. Use of the total difficulties score may ameliorate some of these concerns (internal consistency reliability for the SDQ remains high across numerous studies), as does the comparable findings with the high MHC score. Additionally, while the cut points employed in this study were based on Australian samples in the SEARCH and LSAC cohorts, they were not specifically tested with Aboriginal and Torres Strait Islander children and therefore must be interpreted cautiously [19].

This study presents new information on resilience, gender and wellbeing in Aboriginal and Torres Strait Islander children, with connections to family (and by extension culture and self-identity) proving a particular strength in this cohort. Aboriginal and Torres Strait Islander children predominantly reported positive wellbeing, although boys who scored in the low resilience range appeared particularly vulnerable to emotional/behavioural difficulties. There is an important role for families, schools and communities to identify and strengthen the resilience resources available to Aboriginal and Torres Strait Islander boys, especially in the school and community domains.

## Supporting information

**S1 Checklist.**
(DOCX)

**S1 Table. Aboriginal Families Study sample: Family characteristics by child gender (n = 231).**
(DOCX)

## Acknowledgments

The authors respectfully acknowledge the Aboriginal Custodians of the Lands and Waters of Australia. The authors thank the many Aboriginal and Torres Strait Islander families who have played a role in the Aboriginal Families Study and the Childhood Resilience Study through the community consultations and development of the questionnaires; the women and families that have taken part in these studies; and the many supporting agencies. The authors

would also like to thank current and past members of the Aboriginal Governance Group and members of the fieldwork team for their respective contributions to the study.

## Author Contributions

**Conceptualization:** Deirdre Gartland, Fiona Mensah, Karen Glover, Cathy Leane, Stephanie Janne Brown.

**Data curation:** Deirdre Gartland, Fiona Mensah, Stephanie Janne Brown.

**Formal analysis:** Deirdre Gartland, Fiona Mensah.

**Funding acquisition:** Deirdre Gartland, Fiona Mensah, Karen Glover, Cathy Leane, Stephanie Janne Brown.

**Investigation:** Deirdre Gartland, Arwen Nikolof, Karen Glover, Cathy Leane.

**Methodology:** Deirdre Gartland, Arwen Nikolof, Fiona Mensah, Graham Gee, Karen Glover, Cathy Leane, Stephanie Janne Brown.

**Project administration:** Deirdre Gartland, Arwen Nikolof.

**Supervision:** Deirdre Gartland, Arwen Nikolof, Karen Glover, Cathy Leane, Heather Carter, Stephanie Janne Brown.

**Validation:** Deirdre Gartland, Fiona Mensah.

**Visualization:** Deirdre Gartland.

**Writing – original draft:** Deirdre Gartland.

**Writing – review & editing:** Deirdre Gartland, Arwen Nikolof, Fiona Mensah, Graham Gee, Karen Glover, Cathy Leane, Heather Carter, Stephanie Janne Brown.

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
