## [Decision Letter · Decision Letter 0]

23 Jan 2024

PONE-D-23-26018Resilience and emotional and behavioural wellbeing in boys and girls aged 5-9 years: the Child Resilience Questionnaire in an Australian Aboriginal and Torres Strait Islander cohort study.PLOS ONE

Dear Dr. Gartland,

Thank you for submitting your manuscript to PLOS ONE. After careful consideration, we feel that it has merit but does not fully meet PLOS ONE’s publication criteria as it currently stands. Therefore, we invite you to submit a revised version of the manuscript that addresses the points raised during the review process.

Please find some minor remarks and questions by the reviewers. Please submit your revised manuscript by Mar 08 2024 11:59PM. If you will need more time than this to complete your revisions, please reply to this message or contact the journal office at plosone@plos.org. Please include the following items when submitting your revised manuscript:A rebuttal letter that responds to each point raised by the academic editor and reviewer(s). You should upload this letter as a separate file labeled 'Response to Reviewers'.A marked-up copy of your manuscript that highlights changes made to the original version. You should upload this as a separate file labeled 'Revised Manuscript with Track Changes'.An unmarked version of your revised paper without tracked changes. You should upload this as a separate file labeled 'Manuscript'.If applicable, we recommend that you deposit your laboratory protocols in protocols.io to enhance the reproducibility of your results. Protocols.io assigns your protocol its own identifier (DOI) so that it can be cited independently in the future. For instructions see: https://journals.plos.org/plosone/s/submission-guidelines#loc-laboratory-protocols. Additionally, PLOS ONE offers an option for publishing peer-reviewed Lab Protocol articles, which describe protocols hosted on protocols.io. Read more information on sharing protocols at https://plos.org/protocols?utm_medium=editorial-email&utm_source=authorletters&utm_campaign=protocols.

We look forward to receiving your revised manuscript.

Kind regards,

Inge Roggen, M.D., Ph.D.

Academic Editor

PLOS ONE

Journal Requirements:

3. Please include a complete copy of PLOS’ questionnaire on inclusivity in global research in your revised manuscript. Our policy for research in this area aims to improve transparency in the reporting of research performed outside of researchers’ own country or community. The policy applies to researchers who have travelled to a different country to conduct research, research with Indigenous populations or their lands, and research on cultural artefacts. The questionnaire can also be requested at the journal’s discretion for any other submissions, even if these conditions are not met.  Please find more information on the policy and a link to download a blank copy of the questionnaire here: https://journals.plos.org/plosone/s/best-practices-in-research-reporting. Please upload a completed version of your questionnaire as Supporting Information when you resubmit your manuscript.

Reviewers' comments:

Reviewer's Responses to Questions

**Comments to the Author**

1. Is the manuscript technically sound, and do the data support the conclusions?

Reviewer #1: Yes

Reviewer #2: Yes

2. Has the statistical analysis been performed appropriately and rigorously? 

Reviewer #1: Yes

Reviewer #2: Yes

3. Have the authors made all data underlying the findings in their manuscript fully available?

Reviewer #1: Yes

Reviewer #2: Yes

4. Is the manuscript presented in an intelligible fashion and written in standard English?

Reviewer #1: Yes

Reviewer #2: Yes

5. Review Comments to the Author

Reviewer #1: The concept and study of child resilience is most important particularly as we continue to flesh out the mental health needs of children and young adults. In regions of the world where indigenous populations have faced colonial violence, racism, and dispossession of land and culture, the concept of resilience is critical to understanding the impacts of those kinds of change on child development. I commend the authors for taking on that topic.

I accept the rationale presented by the authors in choosing or developing a definition for resilience. I am curious whether this definition is broadly accepted or if it is more novel? There are three citations provided in support of the definition, but my cursory reading of those papers suggest that the definition blends concepts from those three. I ask because further down, the authors highlight how resilience is sometimes defined in terms of positive functioning. And their definition points to resilience being a process that bridges experience of adversity and wellbeing. So is this defining approach new, and if it is what are the advantages over other approaches?

In the methods section, I have not had the opportunity to use the Tobit model, and so reading its application, and the clear explanation for the choice of this analytic approach has been informative. I assume of course that the data will be made available through PLOSOne on publication? With Figure 1 and 2, I assume the number reported is the scale mean score, and not the upper or lower CI of the estimate? The position of the number is a bit confusing. Perhaps clarify in the figure legend at the bottom?

Finally, in the discussion, the finding that Aboriginal girls scored higher than boys is fascinating to me. The authors put forward the notion that the observed result may be attributed to some social or cultural "scaffolding". Are there other studies into indigenous populations that support this point? Or are the authors basing this on their knowledge and interactions with the Australian indigenous population they are working with?

Thank you for this opportunity to review your work.

Reviewer #2: Thank you for the opportunity to review this paper. I learned a lot from reviewing this research. This was a well done and well written study. I appreciate your attention to detail and your transparency. It made the research easy to follow.

Notes I had:

Are there any psychometrics (e.g. chronbachs’ alpha) that have been previously reported on the measures that were used in this study? If yes, would you be able to report the reliability and validity of these measures?

I was wondering if there were other similar findings in this population or in other populations who may face similar life experiences or who emphasize family, traditions, and self-identity in their children. For example the Inuit in Canada, or those who belong to Native American tribes in America.

These are just somethings that may bolster the external validity of these findings. I can see areas where the findings from your paper may translate to other populations.

6. PLOS authors have the option to publish the peer review history of their article (what does this mean?). If published, this will include your full peer review and any attached files.

Reviewer #1: No

Reviewer #2: No

---

## [Author Response · Author response to Decision Letter 0]

5 Mar 2024

Response to reviewers.

Reviewer #1 

I accept the rationale presented by the authors in choosing or developing a definition for resilience. I am curious whether this definition is broadly accepted or if it is more novel? There are three citations provided in support of the definition, but my cursory reading of those papers suggest that the definition blends concepts from those three. I ask because further down, the authors highlight how resilience is sometimes defined in terms of positive functioning. And their definition points to resilience being a process that bridges experience of adversity and wellbeing. So is this defining approach new, and if it is what are the advantages over other approaches? 

Our definition is the latest approach that is widely accepted but also highlights the ‘sociocultural context’ which can be overlooked by some researchers. We have tried to make this clearer in the text by adding a preamble to our definition.

“These descriptions encompass the two common factors frequently cited as necessary for defining resilience: first, the experience of adversity or stress, and second, the achievement of positive outcomes during or following the exposure to adversity. More recent definitions including the Kickett definition, also encompass the key role of culture, and the potential for growth through adversity. ...”

In the methods section, I have not had the opportunity to use the Tobit model, and so reading its application, and the clear explanation for the choice of this analytic approach has been informative. 

Thank you.

I assume of course that the data will be made available through PLOSOne on publication? 

We are not able to make our data publicly available as the conditions of our ethics approval preclude release of study data. 

Our data sharing statement is as follows: 

Under the conditions of the ethics approval, the Principal Investigator of the Aboriginal Families Study (SB) acts as the custodian for study data and the Intergenerational Health group at Murdoch Children’s Research Institute, Melbourne, Australia, as the co-ordinating unit. Study participants retain ownership of their data and may withdraw consent for their data to be included prior to study outputs. Data sharing via a public repository is precluded under conditions of ethics approval. However, the Aboriginal Families Study Investigator Team and Aboriginal Governance Group welcome inquiries about the data and proposals for collaboration, please contact Lead Investigators, Karen Glover (karen.glover@sahmri.com) and Stephanie Brown (stephanie.brown@mcri.edu.au).

With Figure 1 and 2, I assume the number reported is the scale mean score, and not the upper or lower CI of the estimate? The position of the number is a bit confusing. Perhaps clarify in the figure legend at the bottom? 

Yes, it is the mean that is reported. We have added this into the figure legend at the bottom for Figure 1 & 2 as follows:

e.g. Figure 1. Mean CRQ-P/C scale and domain scores (reported in figure) with 95% confidence interval bars

Finally, in the discussion, the finding that Aboriginal girls scored higher than boys is fascinating to me. The authors put forward the notion that the observed result may be attributed to some social or cultural "scaffolding". Are there other studies into indigenous populations that support this point? Or are the authors basing this on their knowledge and interactions with the Australian indigenous population they are working with?

This came from consultations with our Aboriginal Governance Group and input from our Aboriginal authors. We are suggesting that the boys may require greater scaffolding to build their resilience resources at school from the early school years. There is not much literature available to explore this finding and we have added the following to make this clearer.

“No comparative Australian or international research examining child resilience by gender was identified. A 2009 paper described an analysis conducted to explore gender differences in protective factors using the US Longitudinal Survey of Youth study Waves 3-6 (1986-1996).[37] Notably, the findings apply to youth rather than children, and resilience was defined in terms of not having a negative outcome (rather than as a process as is used in this study). The authors reported that having a positive school environment (in secondary school) was associated with resilience against delinquency and drug use for girls but not for boys. Further research reflecting the current understanding of resilience as a process is imperative.”

Reviewer #2: 

Are there any psychometrics (e.g. cronbachs’ alpha) that have been previously reported on the measures that were used in this study? If yes, would you be able to report the reliability and validity of these measures? 

We have added Cronbach alphas to the measures section as follows:

“Excellent to very good scale reliability has been observed with Cronbach alphas ranging from 0.88 (Connectedness to language) to 0.73 (Family guidance). The Family basic needs scale had good reliability with a Cronbach alpha of 0.61.[17]”

I was wondering if there were other similar findings in this population or in other populations who may face similar life experiences or who emphasize family, traditions, and self-identity in their children. For example the Inuit in Canada, or those who belong to Native American tribes in America.

These are just somethings that may bolster the external validity of these findings. I can see areas where the findings from your paper may translate to other populations. 

This is the first measure of child resilience, and the first developed with Australian Aboriginal community. The only other validated measure of child resilience is the Child and Youth Resilience Measure which was developed with and for ‘general population’ youth and has been primarily used with adolescents. (A validation study of the CYRM with Aboriginal boarding school students indicated the CYRM model of resilience did not work for Aboriginal adolescents). 

Thus findings in other populations tend to be qualitative and difficult to compare. We have tried to better situate our findings in the literature as follows:

“While most of the resilience research in Australia and internationally is focused on youth or adults, the importance of these particular strengths is reflected in the literature. For example, in one interview study, Aboriginal community members identified that the key to child resilience was “knowledge and self-belief that encouraged positive decision making despite challenges”.[24] A recent scoping review of Australian programs, processes, and practices to promote individual and/or collective Indigenous resilience or wellbeing reported that most of the eight publications identified emphasized the need for strategies to strengthen individual or community connection to culture.[12] Finally, a Canadian study utilising Indigenous methodologies reported that strengthening cultural identity and family connections was one of three intersecting processes that facilitate youth resilience and wellness.[25]” 

EDITOR COMMENTS

We have edited our manuscript to match the style requirements.

This option is not available to us - see below.

3. Please include a complete copy of PLOS’ questionnaire on inclusivity in global research in your revised manuscript. Our policy for research in this area aims to improve transparency in the reporting of research performed outside of researchers’ own country or community. The policy applies to researchers who have travelled to a different country to conduct research, research with Indigenous populations or their lands, and research on cultural artefacts. The questionnaire can also be requested at the journal’s discretion for any other submissions, even if these conditions are not met. Please find more information on the policy and a link to download a blank copy of the questionnaire here: https://journals.plos.org/plosone/s/best-practices-in-research-reporting. Please upload a completed version of your questionnaire as Supporting Information when you resubmit your manuscript.

We have completed and uploaded.

We have included a data availability statement in the submission form including a reason for why we are unable to make your data freely accessible.

Ethics is included in the methods only.

We have included a caption for out S1 Table.

---

## [Editor Report · Decision Letter 1]

20 Mar 2024

The Childhood Resilience Study: Resilience and emotional and behavioural wellbeing experienced by Australian Aboriginal and Torres Strait Islander boys and girls aged 5-9 years.

PONE-D-23-26018R1

Dear Dr. Gartland,

We’re pleased to inform you that your manuscript has been judged scientifically suitable for publication and will be formally accepted for publication once it meets all outstanding technical requirements.

Kind regards,

Inge Roggen, M.D., Ph.D.

Academic Editor

PLOS ONE
---

## [Editor Report · Acceptance letter]

5 Apr 2024

PONE-D-23-26018R1 

PLOS ONE

Dear Dr. Gartland, 

I'm pleased to inform you that your manuscript has been deemed suitable for publication in PLOS ONE. Congratulations! Your manuscript is now being handed over to our production team.

Kind regards, 

on behalf of

Prof. Inge Roggen 

Academic Editor

PLOS ONE